# Land-use intensity of electricity production and tomorrow's energy landscape

**Jessica Lovering**[1]*, **Marian Swain**[2], **Linus Blomqvist**[3], **Rebecca R. Hernandez**[4,5,6,7]

**1** Fastest Path to Zero Initiative, University of Michigan, Ann Arbor, MI, United States of America, **2** Breakthrough Institute, Oakland, CA, United States of America, **3** Bren School of Environmental Science & Management, University of California, Santa Barbara, CA, United States of America, **4** Energy and Resources Group, University of California, Berkeley, California, United States of America, **5** Earth Sciences Division, Climate and Carbon Sciences Program, Lawrence Berkeley National Laboratory, Berkeley, California, United States of America, **6** Department of Land, Air, & Water Resources, University of California, Davis, Davis, CA, United States of America, **7** Wild Energy Initiative, John Muir Institute of the Environment, University of California, Davis, Davis, CA, United States of America

* lovering@umich.edu

**Data Availability Statement:** All relevant data are within the paper and its Supporting Information files.

**Funding:** R.R.H. received funding for this project from the USDA National Institute of Food and

## Abstract

The global energy system has a relatively small land footprint at present, comprising just 0.4% of ice-free land. This pales in comparison to agricultural land use– 30–38% of ice-free land–yet future low-carbon energy systems that shift to more extensive technologies could dramatically alter landscapes around the globe. The challenge is more acute given the projected doubling of global energy consumption by 2050 and widespread electrification of transportation and industry. Yet unlike greenhouse gas emissions, land use intensity of energy has been rarely studied in a rigorous way. Here we calculate land-use intensity of energy (LUIE) for real-world sites across all major sources of electricity, integrating data from published literature, databases, and original data collection. We find a range of LUIE that span four orders of magnitude, from nuclear with 7.1 ha/TWh/y to dedicated biomass at 58,000 ha/TWh/y. By applying these LUIE results to the future electricity portfolios of ten energy scenarios, we conclude that land use could become a significant constraint on deep decarbonization of the power system, yet low-carbon, land-efficient options are available.

## 1. Introduction

Providing energy for a population of a projected 10 billion by mid-century has many impacts on public health and the environment beyond just carbon emissions. These impacts include water use, materials consumption, local particulate pollution, and land use. The land footprint of energy systems can displace natural ecosystems, lead to land degradation, and create trade-offs for food production, urban development, and conservation. For example, a recent analysis showed that energy sprawl is now the largest driver of land-use change in the United States [1]. Furthermore, land-intensive energy sources may face growing opposition during the siting process, potentially slowing down the rate of the clean energy transition. Giordono et al. (2018) analyzed proposed wind farms in the western United States and found that over 35% of projects faced some form of opposition [2]. The land footprint of energy may become an even

Agriculture Hatch Project 1010512, the Department of Land, Air and Water Resources at UC Davis, and the UC President's Postdoctoral Fellowship. https://nifa.usda.gov/grants https://www.lawr.ucdavis.edu/ https://ppfp.ucop.edu/info/ The funders had no role in study design, data collection and analysis, decision to publish, or preparation of the manuscript.

**Competing interests:** The authors have declared that no competing interests exist.

larger driver of environmental impacts in the coming decades, if energy consumption rises rapidly in emerging economies and countries shift their mix of energy sources to meet decarbonization targets [3], potentially towards more land-intensive energy sources.

The land footprint of energy is seldom considered in regional and global assessments of decarbonization pathways, land-use change, and biodiversity threats, with the occasional exception of particularly land-intensive sources like bioenergy [4–10]. There is a need to consider land use as a key factor in energy systems planning, along with other environmental impacts, public health, greenhouse gas emissions, affordability, and energy security.

There are only a limited set of existing studies that assess land-use intensity of energy (LUIE) across all major electricity sources and all have methodological weaknesses. Previous studies calculated LUIE based on a single installation [11], a small number of non-randomly selected facilities [12–14], or use modeled electricity generation data. LUIE figures drawing only on a single or handful of sample units may by chance represent unusually small or large facilities, and thus misrepresent the full distribution of intensities, which are better characterized by the mean or median of a larger sample. Furthermore, modeled electricity generation data may not reflect actual performance. As such, LUIE figures drawing on larger, real-world samples are needed.

In this study, we collected and calculated the land-use intensity (measured as hectares occupied per terawatt-hour of electricity generated in a given year [ha/TWh/y]) for real-world electricity generation–not hypothetical or modeled electricity generation–across all major sources of electricity and a broad geographic distribution. We focus on the land footprint of electricity only, as most future energy scenarios predict disproportionate growth in electricity consumption as transportation and industry electrify to reduce emissions, and electricity production has the broadest range of technologies with diverse land-use impacts [15, 16]. Our data set covers 73 countries and 45 US states. Data are collected from published studies as well as public records, datasets, and original geospatial analysis (see Supporting information for full details). We cover coal, natural gas, nuclear, wind, solar photovoltaic (PV), concentrated solar power (CSP), geothermal, hydroelectric, and biomass (including electricity from dedicated biomass feedstock production, hereafter called "dedicated biomass"; and electricity from waste and residue biomass, hereafter called "residue biomass").

We apply our LUIE results to ten prominent scenarios for future energy supply. These scenarios vary greatly in their mix of renewables, fossil fuels, and nuclear energy, but all had large increases in global electricity generation. Our study is the first to reflect the diversity of land-use intensity both within and across energy technologies as they operate in the real world. This new dataset will aid policy-makers in a quantitative and comparative understanding of the balance between energy, land, and climate change mitigation and the implications of a build-out of low-carbon electricity sources on global land use.

## 2. Methods

Our LUIE dataset is compiled from ten peer-reviewed studies, eight published reports from government agencies and national labs, nine databases, and two original geospatial analyses. To provide LUIE results representative of the current state of each energy technology, we required that data sources represent existing, operational energy facilities and real world, rather than modeled, electricity generation data. The one exception is for concentrating solar power (CSP), as we could only find real generating data for a handful sites; here, we also included a dataset with a range of expected generation from existing CSP sites [17]. As we only focused on electricity generation, we excluded liquid biofuels used in transportation and traditional biomass used directly for heating and lighting.

We drew on peer-reviewed literature to aggregate data for coal, natural gas, and biomass LUIE. For geothermal, hydroelectric, and solar, we combined data from past studies and

publicly available datasets. For wind and nuclear, we calculated area requirements using original measurements in Google Earth Pro and electricity generation data from the US Energy Information Administration (EIA) databases. Where possible, we obtained globally representative samples of energy facilities, but due to limitations on electricity generation data for individual power plants, data for the following energy sources include only facilities in the United States: nuclear, wind, and ground-mounted PV. For solar PV and wind, we expect LUIE to be similar across countries as the technology is produced by a small number of international suppliers and depends mostly on solar insolation. For nuclear power, we expect the LUIE based on US plants to be an upper bound, as most other countries with large nuclear fleets have higher numbers of reactors at each site, leading to economies of scale in terms of occupied land.

Our LUIE calculations do not include land that is occupied by the upstream manufacturing of electricity generating facilities (e.g., the land required to mine materials for solar panel or wind turbine production, or the materials that go into nuclear or coal power plants). We also exclude land required for electricity transmission infrastructure (e.g., high voltage transmission corridors), offshore area impacts (for wind farms and natural gas drilling), and underground impacts for geothermal, natural gas, and coal mining).

The formula to measure direct LUIE (Eq 1), involves dividing the land occupied by an electricity-producing facility by the energy it produces over a year [13, 18–22]. For most combustion-based generation—except nuclear—the power plant is only a small proportion of the land occupied to produce energy, with fuel production taking up a much larger amount of land. We call the area for fuel production indirect land use (Eq 2). This indirect land use applies to coal, natural gas, dedicated biomass, and nuclear, which require externally-sourced fuel. Total LUIE (Eq 3) is the sum of direct and indirect LUIE. Where data for a single facility was incomplete, for example only direct LUIE was provided, it was combined with the average indirect LUIE result from other sources to calculate total LUIE.

$$LUIE_{direct} = \frac{A_{direct}}{Energy} \left[\frac{ha * y}{TWh}\right]$$

**Eq 1:** Direct land use intensity

$$LUIE_{indirect} = \frac{A_{indirect}}{Energy} \left[\frac{ha * y}{TWh}\right]$$

**Eq 2:** Indirect land use intensity (applicable energy systems: coal, natural gas, biomass, nuclear)

$$LUIE_{total} = \frac{A_{direct} + A_{indirect}}{Energy} \left[\frac{ha * y}{TWh}\right]$$

**Eq 3:** Total land use intensity

$$A_{footprint} = \text{Footprint } [ha]$$
$$OR$$
$$A_{spacing} = \text{Footprint } [ha] + \text{Spacing } [ha]$$

**Eq 4:** Direct area definitions (applicable energy systems: natural gas, wind)

For two electricity sources (natural gas and wind), we offer two definitions of occupied land for our calculation of land use intensity: "footprint" and "spacing" area (Eq 4). Footprint area represents land directly covered by infrastructure, while spacing area is the entire area within the perimeter of a production site (further details in S1 Text). For each electricity source, we included all individual LUIE values and calculated the median, average, standard deviation, and interquartile range. To determine if our calculated LUIEs were statistically distinguishable, we performed an ANOVA with Tukey's pairwise comparison. We anticipate large variability in LUIE within and across energy technologies, as has been demonstrated and discussed by previous studies [23–26].

Details on our data sourcing for each technology are provided below. A table summarizing the characteristics of the data in each source is provided in the S1 Table.

**Coal:** Total LUIE for coal (n = 30) includes direct land impacts from power plant infrastructure and indirect impacts from coal mining, processing, and transportation for the US and Canada. Several studies performed case studies or lifecycle analysis -both direct and indirect land use—on a small number of coal power plants with varying technologies, including mining and transportation of fuel and waste disposal; these included Fthenakis & Kim (2009), Hertwich et al. (2015), Spitzley & Keoleian (2005), Smil (2010), and Gates (1985) [12, 27–31]. Indirect land use, specifically from mining, is the dominant contributor to coal's land-use intensity, and thus a broader survey that focuses on land occupation for regional coal mining can give a more comprehensive estimate. Jordaan (2010) and McDonald et al. (2009) perform surveys of the land used for coal mining in Canada and the US, respectively [13, 32]. These two studies of indirect land-use were given in units of embodied energy for the mined coal, so we applied a 35% conversion efficiency to convert to electricity units, based on Ftheankis & Kim (2009) [12]. This gives an estimate of hectares per TWh for just the indirect, mining component of coal's land use intensity.

**Natural Gas:** Total LUIE for natural gas (n = 17) includes direct impacts from power plant infrastructure and indirect impacts from natural gas drilling and transportation infrastructure. Footprint LUIE represents the area covered by gas well pads, access roads, and pipelines. Five sources provided data on footprint LUIE [28, 32–35]. Spacing LUIE refers to the entire production field, including all the area in between well pads, even if that land does not have any structures or roads covering it. The US National Energy Technology Laboratory (2014) and US Department of Energy (1983) complete detailed life-cycle assessments for both direct and indirect land-use, including extraction, purification, pipeline transmission, and power plant [33, 34]. Spitzley & Keoleian (2005) and Smil (2010) assess direct land-use through case studies of various natural gas power plants technologies [28, 30]. Jordaan (2010) and Bryce (2011) provided figures only for indirect impacts from natural gas drilling for Canada and the US [32, 35]. Jordaan et al. (2017) calculates lifecycle land use intensity for natural gas, from wells, to pipelines, to power plants [24]. McDonald et al. (2009) and Copeland et al. (2011) provided calculations for indirect spacing LUIE, assessing the area fragmented by natural gas drilling and pipelines [13, 36].

**Nuclear:** Land use for nuclear includes direct impacts (n = 59) from the power plant and indirect impacts from the uranium fuel cycle, including mining, milling, conversion, enrichment, and fabrication. We collected original data for direct land-use for all operating nuclear power plants in the United States, by drawing polygons around each power plant using Google Earth Pro. EIA provides data on each plant's electricity output [37]. Finch (1997), Eliasson & Lee (2003), Harries et al. (1997), and Schneider (2013) survey land area for uranium mining and processing [38–42], mostly in Australia, which averages 0.08 ha/TWh/y when we converted these per ton measurements into electricity units. Fthenakis & Kim (2009) was the only study that provides an estimate for the other aspects of the nuclear fuel cycle: conversion,

enrichment, and fabrication [12]. This is a constraint on data availability, as Fthenakis & Kim (2009) looked only at uranium mining in the US, where almost no uranium mining occurs today. In the US, spent fuel is stored on-site and is therefore included in our direct LUIE calculation. Although the back-end of the nuclear fuel cycle is undetermined in the U.S., we did find two studies that estimated land use for the now-cancelled Yucca Mountain waste repository, which would add an additional 0.012–2.9 ha/TWh to the total LUIE for nuclear[12, 43]. Countries that recycle or reprocess their spent fuel will likely have less land occupation.

We also estimated the additional land-use occupied by exclusion zones around the two major nuclear power accidents at Chernobyl in Ukraine (260,000 ha) [44], and Fukushima in Japan (63,000 ha) [45]. We calculated the LUIE of nuclear accidents by combining these two exclusion zones and dividing that area by total historical nuclear power generation (~82,000 TWh) [46], which resulted in an additional LUIE of 3.9 ha/TWh/y. However, in both cases, the exclusion zones are at least partially inhabited and, in the case of Chernobyl, the zone is occupied by abundant wildlife [47].

**Hydroelectric:** The direct area of hydroelectric dams is the area flooded by the reservoir. Our dataset (n = 962) is compiled from International Commission on Large Dam's (ICOLD) World Register of Dams database and represents single-use hydroelectric dams in eighty countries [48]. The World Register of Dams provided data on mean annual electricity and reservoir area. We exclude run-of-the-river hydroelectric projects since they represent a small portion (roughly 4%) of worldwide hydroelectric capacity and reliable generation data could not be found [49]. However, results from Fthenakis & Kim (2009) suggest LUIE for run-of-the-river projects are much smaller than for traditional hydroelectric (about 10 ha/TWh/y) [12].

**Biomass:** Like other combustibles, the land impacts from biomass include the direct area of the power plant as well as the area needed to supply the feedstock for the plant (indirect LUIE). Our dataset for dedicated biomass (n = 14) represents woody biomass production from willow, poplar, and spruce trees. Data are drawn from six sources [12–14, 28, 50, 51]. For residue biomass, we assume no land requirement for feedstock production. Spitzley & Keoleian (2005), Fthenakis & Kim (2009), Kumar et al. (2003), and Smil (2010) provide generation and direct area information for various biomass plants [12, 28, 36, 51]. Coal power plants can also be used as a proxy since it is common to retrofit a coal plant to burn biomass, but we would expect a biomass plant to have a larger LUIE since the plant runs at lower efficiency. Dijkman & Benders (2010), Kumar et al. (2003), and McDonald et al. (2009) calculate indirect land-use for biomass feedstock production looking at different crops [13, 50, 51].

**Wind:** While there are several studies of power density for wind farms (m$^2$/MW), we created an original dataset to calculate the land use intensity of existing wind farms using historic electricity generation data. Land impacts from wind come from the area covered by wind turbines and access roads. We calculate both footprint and spacing LUIE results for wind (n = 57). Footprint area represents only the area physically covered by the turbine pad and access roads; spacing area includes all the area in between turbines. Our dataset is generated from a randomized sample of operating US wind farms over 20 MW from EIA. We used EIA Form 860 and Form 923 to gather data on installed capacity and annual electricity output for each wind farm for 2013 [52, 53]. We combined this with measurements of the footprint and spacing area of each wind farm calculated using Google Earth Pro. For footprint area, we traced perimeters around each turbine pad and the access roads connecting them. For spacing area, we traced the perimeter of the entire wind farm, including all the space in between turbines.

**Solar PV:** Our datasets for ground-mounted PV are based on existing, operational plants over 20 MW in 18 US states with capacity factors over 5%. For all sites, annual electricity generation data came from EIA Form 923 [53]. Area measurements came from Hernandez et al.

(n = 17) [54]; Ong et al. (n = 61) [21]; and the Solar Energy Industries Association (n = 13) [55]. For ground-mounted PV, we define direct area as the area of panels or heliostats, roads established during development, and all ancillary facilities. Ancillary facilities may include new service roads, power collection systems, communication cables, overhead and underground transmission lines, electrical sites, switchyards, project substations, meteorological towers, thermal storage units, and operations and maintenance facilities. Integrated Solar PV, i.e. roof-top solar, is assigned an LUIE of zero in this study.

**Solar CSP:** There are very few operating solar CSP plants globally for which we could find real electricity generation data. Ong et al. (2013) and Lilliestam et al. (2021) provide data on land area for solar CSP plants, but both provide only estimated or anticipated electricity generation. Ong uses two different capacity factors to provide a range of generation for each plant. We found real electricity generation data for ten plants, primarily in the US with one in Italy and two in Spain. Because solar CSP could play a large role in future energy systems, we decided to include LUIE from datasets that use estimates of electricity generation. This resulted in n = 101 data points for unique power plants across 13 countries. However, for plants where there was overlap across the datasets, and we had real electricity generation, we found that the larger estimated dataset over-estimated electricity generation by ~30%, which would bias the LUIE figure downward.

**Geothermal:** Geothermal land impacts include the area covered by power plant infrastructure and injection wells. Bertani (2005) provided a detailed list of worldwide geothermal power plants; however, their measured areas represented the entire expanse of the underground geothermal reservoir, only a fraction of which had aboveground land disturbance from the power plant and production wells [56]. We cross-referenced Bertani's generation data with land use data from geospatial measurements from the Global Energy Observatory (GEO) online database [57]. Bertani's land area measurements were ~60 times larger than those measured from satellite imagery in the GEO database. Our resulting dataset included 26 plants in 18 countries.

## 2.1 Application to scenarios

The ten scenarios we assess are all global decarbonization pathways that make normative choices about electricity demand, electrification rates, and generating technologies (see S1 Text). The exception is IEA's 6 degree scenario, which is a "business-as-usual" forecast. These scenarios vary in their assumptions about total electricity demand and the technology mix (see S1 Fig in S1 Text), as well as the end year of their projections (the JD scenario is for the year 2030 [58], Brook is for 2060 [59], and all others are for 2050). They were not selected based on economic or technical feasibility, but rather to represent a diverse range of future electricity scenarios, illustrating the possible land use implications of different decarbonization pathways. To determine the total land area required for electricity generation in the current (2017) and future scenarios, we multiplied the average LUIE result for each energy technology (in ha/TWh/y) by the amount of generation from that technology (in TWh/y) and summed totals over all electricity sources. Using the average LUIE provides a more accurate land use estimate when summing up over many sources, as using the median tends to underestimate total land use when multiplied by all electricity consumption.

A thorough study of the land use implication for fossil- or biomass-fueled power plants with Carbon Capture and Storage (CCS) has yet to be performed, although data from Hertwich et al. suggest it would increase footprint by 40% compared to a plant without CCS. If a scenario included fossil or biomass generation with CCS, we multiplied our natural gas, coal,

and biomass LUIEs by 1.4 [27]. Electricity generation from oil combustion was included in some scenarios in very small quantities; we used the footprint LUIE from a natural gas plant for this figure, as estimates in the literature are not available.

To understand the significance of the differences across all future scenarios, we also propagated the errors (standard error of each electricity source) through the total land use calculation to provide an uncertainty range for each scenario's total land use. In all scenarios, the uncertainty is dominated by the standard deviation in the LUIE of hydroelectric, which is large due to the regional variability of hydroelectric resources.

## 3. Results

Our LUIE calculations include land occupied by the electricity-producing facility (called "direct area") and, if applicable, the land needed to source power plant fuel (called "indirect area"). For wind and natural gas, we offer two definitions of occupied land: "footprint" and "spacing" area. Footprint land is physically occupied by components of the power plant or fuel extraction equipment, while spacing is the land in between physical components in an electricity generation or fuel extraction site. For wind, footprint area measures only the area covered by turbine pads and access roads, while spacing area measures the entire area within the boundaries of the wind farm. For natural gas, footprint area for the indirect land use measures only the area covered by well pads, access roads, and pipelines, while spacing area includes the entire area inside the perimeter of a natural gas production field.

We find that median LUIE varies by four orders of magnitude across the electricity sources considered in this study (Fig 1, Table 1). Nuclear had the lowest median LUIE at 7.1 ha/TWh/y, and dedicated biomass the highest at 58,000 ha/TWh/y.

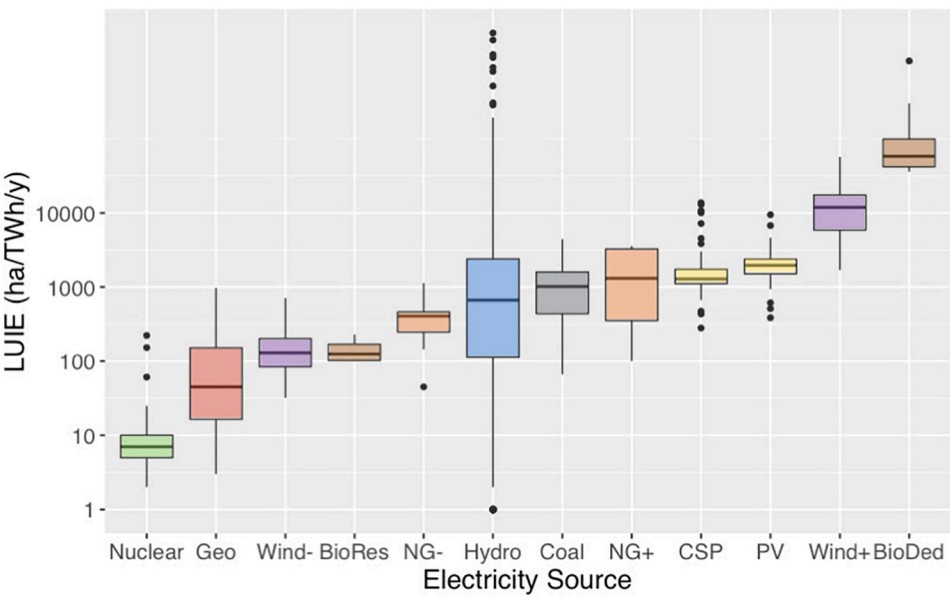

**Fig 1. Land use intensity of electricity (LUIE: ha/TWh/y), shown on log scale.** Boxes represent the inter-quartile range with the median as the middle bar. Whiskers extend to the highest or lowest data point that is within 1.5 times the inter-quartile range; points outside this range represent outliers. Electricity sources: nuclear energy (Nuclear), geothermal energy (Geo), wind energy with footprint only, (Wind-), natural gas footprint only (NG-) and including spacing (NG+), hydroelectric power for single purpose dams (Hydro), coal (Coal), concentrating solar power (CSP), ground-mounted photovoltaic solar energy (PV), wind energy with footprint (Wind-) and spacing (Wind+), and residual biomass (BioRes) and dedicated biomass (BioDed).

**Table 1. Land use intensity of electricity (LUIE) showing total direct and indirect land use (ha/TWh/y).** We show median, mean, and interquartile range (IQR) for the LUIE, along with the number (n) of observations for each power source. We performed an ANOVA analysis with Tukey's pairwise comparisons on the $\log_{10}$ of LUIE for different sources, which is represented by different letters. Sources that share a letter have mean LUIE that are not statistically different. Hydroelectric was excluded from the ANOVA analysis because its variance was too large.

| | ANOVA Tukey's Pairwise | LUIE Median | LUIE IQR | LUIE Mean | LUIE Standard Error | LUIE n |
|---|---|---|---|---|---|---|
| **Nuclear** | A | 7.1 | 4.8 | 15 | 4.4 | 59 |
| **Geothermal** | B | 45 | 150 | 140 | 46 | 26 |
| **Wind (footprint)** | C | 130 | 120 | 170 | 18 | 57 |
| **Residue biomass** | C D | 130 | 71 | 150 | 31 | 4 |
| **Natural gas (footprint)** | C D | 410 | 210 | 410 | 58 | 17 |
| **Hydroelectric (single purpose dams)** | — | 650 | 2,300 | 15,000 | 4,300 | 952 |
| **Coal** | D | 1,000 | 1,200 | 1,100 | 170 | 30 |
| **Solar CSP** | D E | 1,300 | 650 | 2,000 | 220 | 101 |
| **Natural gas (spacing)** | D E | 1,900 | 2,800 | 1,900 | 890 | 4 |
| **Ground-mounted PV** | E | 2,000 | 860 | 2,100 | 120 | 94 |
| **Wind (spacing)** | F | 12,000 | 12,000 | 15,000 | 1,700 | 57 |
| **Dedicated biomass** | G | 58,000 | 59,000 | 160,000 | 77,000 | 14 |

Indirect land use for combustion-based electricity–land used for fuel sourcing for coal, natural gas, and biomass—is a larger share of LUIE than direct land use. Indirect land use comprises over 90% of total land use for natural gas generation, approximately 55% for coal generation, and over 99% for dedicated biomass (see S1 Text for more details on data sources). The opposite is true for nuclear power, where indirect land use for uranium mining is only 10% of total LUIE and the majority of land impacts come from the power plant itself. When including accident exclusion zones in the total LUIE for nuclear, indirect land use drops to 6% of total. Although our calculations do not include upstream land impacts from manufacturing of materials, other studies of renewable energy technologies find upstream (indirect) land demands to be negligible, less than 1% of total land use [12].

To test for statistical differences in LUIE across different sources, we conducted an ANOVA analysis with Tukey's pairwise comparisons on the natural logarithm of the means of the different sources (Table 1). The ANOVA model uses pooled variance and has an $R^2$ of 90.46%. Hydroelectric was excluded due to its large variance, which would compromise all the pairwise comparisons since it increases the pooled variance.

According to this analysis, dedicated biomass, wind (footprint and spacing), geothermal, and nuclear were significantly different from every other source. Ground-mounted PV, solar CSP, and natural gas (spacing) were not significantly different from each other; the same was true for solar CSP, natural gas (spacing), and coal. Natural gas (spacing) was not significantly different from natural gas (footprint). Hydroelectric has a large variance, even after we narrowed our analysis to dams that are only used for power generation, excluding dams with secondary purposes for irrigation, flood control, and drinking water supply.

## 3.1 Comparing LUIE and life-cycle GHG emissions

Land-use intensity and GHG emissions are both important metrics for assessing the environmental impacts of electricity production. We identify several electricity-generating technologies that minimize both land use (from our LUIE results) and GHG emissions (median results for the entire electricity life-cycle from IPCC) [60], including rooftop PV, nuclear, wind (footprint only), and geothermal (Fig 2). The large variance of hydroelectric, biomass, and geothermal reflect the dependence of these sources on local conditions. A dam in a steep mountain

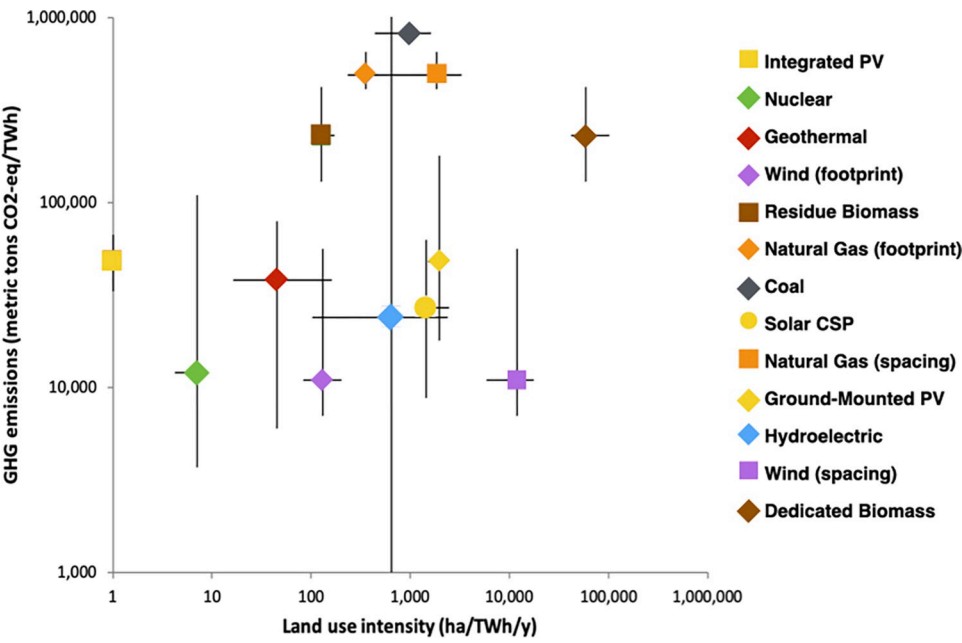

**Fig 2. Relationship between the land use intensity of electricity (ha/TWh/y) and lifecycle GHG emissions (metric tons $CO^2$-eq/TWh) on a log scale.** Error bars represent interquartile range. GHG emissions source data: IPCC Fifth Assessment, Working Group III [60].

valley generates large amounts of electricity on very little land, compared with a dam in a shallow basin. Similarly, the type of land flooded to create the reservoir, or the type of biomass feedstock used can lead the large difference in lifecycle GHG emissions.

## 3.2 Future energy scenarios

We applied our mean LUIE results to the electricity mix of future scenarios for the global power sector, as well as to today's global electricity mix [61], to determine the current and projected land requirements for future global electricity roadmaps (Fig 3). Our LUIE results suggest that current total global land use for electricity production is approximately 72 (±1.7) Mha, with 80% of that land used for hydroelectric dams.

We assessed ten global decarbonization pathways from six different organizations and studies: the 2, 4, and 6 degree Celsius scenarios from the International Energy Agency's *Energy Technology Perspectives* (hereafter "IEA") [62], Greenpeace's *Energy [R]evolution* ("GP") [63], World Wildlife Fund's *Energy Report* ("WWF") [64], three scenarios from the *Global Energy Assessment* ("GEA") [10], Jacobson & Delucchi ("JD") [58], and Barry Brook ("Brook") [59]. Real-world land requirements vary by region and the dynamics of land-use change are highly context-dependent. These projections are not intended as forecasts, but rather as estimates of the scale of land use that would be needed for electricity production in hypothetical decarbonized electricity portfolios.

Our analysis suggests the possibility of a significant expansion of the land footprint for electricity in the coming decades, ranging from an additional 30–80 Mha for physical footprint to and additional 80–800 Mha when spacing is included. The scenario with the lowest total land-use was the IEA 6 Degree scenario, which is a business-as-usual scenario that includes a large share of fossil fuels. The WWF and Greenpeace scenarios also had low total land use, but this was in part due to their lower overall projected electricity consumption, as well as their limited

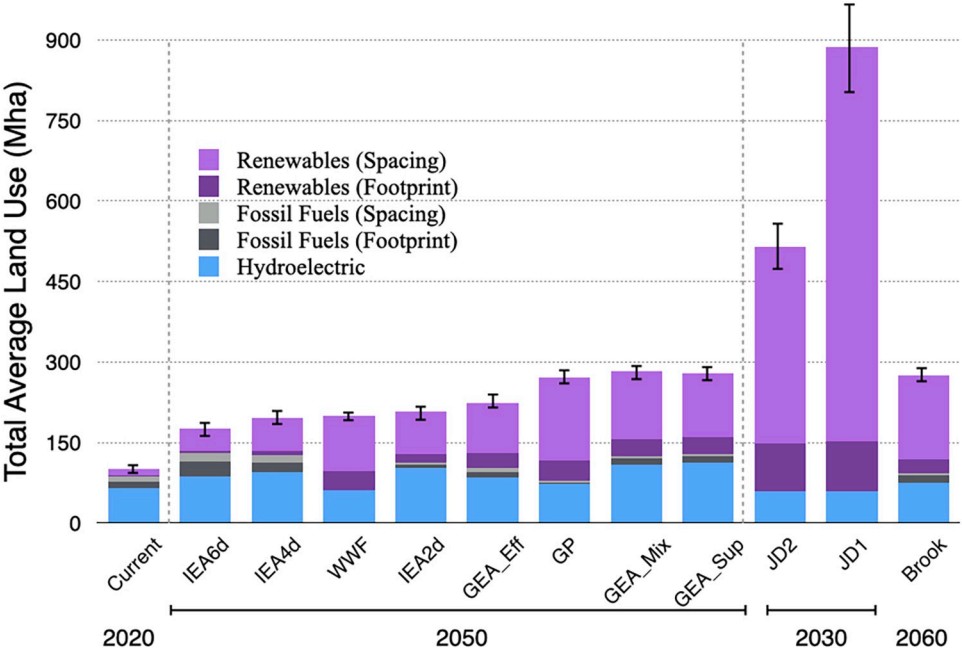

**Fig 3. Land area (Mha) for future electricity generation scenarios, broken down by source of land use: Hydroelectric, fossil fuels, non-hydro renewables, and spacing from wind and natural gas.** Land use for biomass electricity is included in non-hydro renewables, but we assume all biomass comes from residue or waste for these calculations, thus representing a lower bound. JD1 refers to the Jacobson & Delucchi scenario assuming all wind is onshore, and JD2 assumes 50% of wind is onshore and 50% is offshore. Total land required to generate electricity in each future decarbonization scenario is shown with standard errors. GEA_Sup, GEA_Eff, and GEA_Mix are the GEA Supply, Efficiency, and Mixed scenarios respectively. Electricity generation data for the current mix (2017) comes from the BP Statistical Review.

reliance on large hydroelectric. Brook had lower land-use despite higher overall electricity consumption, primarily due to their reliance on nuclear power, which has the lowest LUIE. The Jacobson scenarios had the highest land use both because they were converting all global energy use to electricity, and they also rely extensively on wind and solar.

The projected expansion of land-use across these scenarios is a similar order of magnitude to the value projected for global urban expansion (60–241 Mha) [65], and when spacing is included this may exceed forecasted cropland expansion (average 160–320 Mha of various projections) [6]. If biomass was to come from dedicated feedstocks, the additional land required would be between 80 and 700 Mha across these scenarios. For comparison, Jacobson et al. (2017) estimated that the land required for a 100% renewable system would be lower than our calculation (35 Mha or 177 Mha with spacing), but their land-use figures represent hypothetical electricity generation, which tends to be lower than realized generation from our surveys [16]. Trainor et al. (2016) calculated additional land use from EIA scenarios in the US and found land use could grow by 18–24 Mha by 2040, but this is for all energy supplied in the US (not just electricity) [1].

## 3.3 Sourcing of biomass carries great uncertainty

Future biomass demand will likely be met by a mixture of waste or residues and dedicated feedstocks. However, the average land-use intensity of residue and dedicated biomass differs by four orders of magnitude. To represent an upper boundary on our results, we could assume all biomass comes from dedicated feedstock production. This upper bound estimate results in

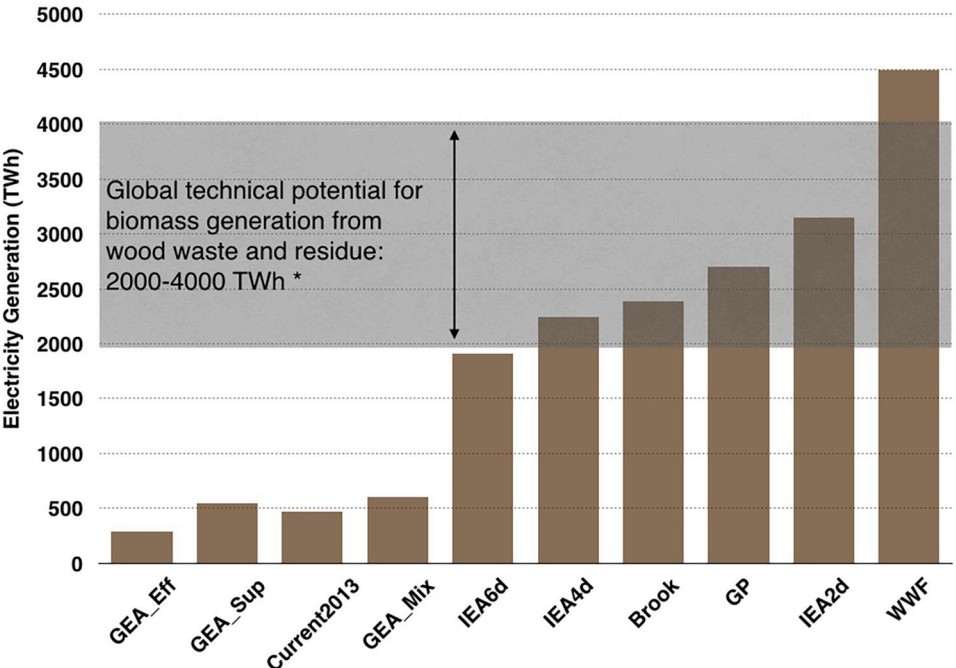

**Fig 4. Amount of electricity sourced from biomass in each of the scenarios we evaluated.** Most scenarios do not specify whether the biomass will be sourced from dedicated crops or managed forests, or sourced from waste and residue. However, several scenarios include more biomass combustion than could be reasonably sourced from waste and residues, assuming all waste and residue produced globally could be economically collected. *Global technical potential for biomass production comes from Searchinger and Heimlich (2015).

biomass comprising over 99% of the total land use in future energy scenarios (unless the scenario excludes biomass). The GP, WWF, and GEA energy scenarios reviewed here specify that the biomass in their scenarios should come only from forestry and agricultural wastes and residues, rather than dedicated production. The level of biomass required in those scenarios is within the range of global technical potential [66], see Fig 4 but estimates of global technical potential do not reflect economic or geographic constraints on biomass residue recovery (see S1 Text). There is also evidence at the regional level that residues alone are unlikely to meet bioenergy demand, which could result in increased logging and displacement of other wood products [67]. To take a lower bound on biomass, we could assume all feedstock comes from waste or residue. Then biomass constitutes only about 1% of total land use in future energy scenarios.

When median LUIE results are applied to global decarbonization scenarios, land use for the power sector could grow by a doubling to an order of magnitude if spacing is included. See S1 Text for full details on how we applied our LUIE to a range of future energy scenarios. However, if the entire energy system is decarbonized, including transportation and industry, global electricity demand could grow by 3–5 times, thus implying an even larger impact on total land use.

## 4. Discussion

Renewable energy sources like ground-mounted PV, CSP, and wind feature prominently in many decarbonization scenarios, but since they can have higher land use intensity than fossil fuels, large-scale deployment of these technologies could considerably increase energy sprawl and loss of natural habitat. The types of landscapes impacted will vary by energy source, and

while there are several opportunities for mitigating the land requirement of low-carbon electricity systems, there is also evidence that renewable energy development to date has often occurred on previously undeveloped land [54, 68].

Some power technologies can produce electricity without requiring additional land. Solar PV can be placed on rooftops, over parking lots, or on top of inland water body surfaces, known as floatovoltaics [69, 70]; PV and wind turbines can be built on degraded, contaminated, or on top of agricultural land—the latter with PV, known as an agrivoltaic systems [71, 72]; biomass feedstock can be sourced from residues and waste materials [66]; and nuclear power plants (as well as wind and PV) can be built offshore [73–75]. Dams that were originally constructed for water supply, irrigation, or flood control can be retrofitted with hydroelectric capabilities [76]. However, there are limitations on scaling these non-additional sources. Integrated PV faces barriers owing to economic, policy, and technological constraints. A recent estimate put the technical potential of rooftop-mounted solar PV in the United States at 1,400 TWh/y–about 38% of current US electricity demand [77]. As a technical potential, this estimate is higher than the economic or market potential for the technology. Currently, only about one-third of US solar capacity is in distributed rooftop installations, while the rest is from ground-mounted, utility-scale power plants [78], of which, in the case of California, the plurality are sited in natural habitats like scrublands and shrublands [54]. Technological advances that improve efficiency and/or electricity generation could arguably reduce LUIE but evidence of a causal relationship is lacking. For example, the manufacturing and construction of bifacial PV (light collected from both sides of the panel surface) is driven by estimates of 20–40% electricity gains over monofacial panels, which could ostensibly drive smaller power plant footprints to meet finite electricity needs. However, the exponential increases in electricity demand globally and the nascent state of the technology's economic and reliability outcomes mean that impacts on real-world LUIE are yet to be determined [79].

Denholm et al. (2009) estimate that half of US wind is co-sited with cropland or pasture [68]. Additionally, areas with good wind resources and proximity to end users do not always overlap with existing agricultural area, sometimes requiring wind energy development on previously undeveloped land, as was recently found to be the case in California [80]. Finally, energy infrastructure can create habitat fragmentation and disturbance that adversely affects wildlife behavior within and beyond the boundary of the physical footprint [7, 37, 81–83]. Studies seeking to map existing and future energy infrastructure (e.g., Jenkins et al., 2021) may, in part, inform decisions regarding impacts on wildlife; however, projections including spatially-explicit footprints require robust, accurate, and representative model parameters[84].

In contrast to previously published studies which base their LUIE figures on either a single power plant or engineering calculations with hypothetical parameters, our LUIE estimates draw on large samples of real-world electricity generation. Other published figures are often far from our estimated means or medians. For example, Fthenakis and Kim's figure for hydro, drawing on a single reservoir from Colorado, is equivalent to about 400 ha/TWh/y, and the central estimate provided by McDonald et al. (2009) is equivalent to 5,400 ha/TWh/y. This can be compared with our sample of 952 hydroelectric installations, which has a median of 650 and a mean of 15000. The land-use intensities for nuclear power are 13 ha/TWh/y in Fthenakis and Kim, and 240 ha/TWh/y in McDonald et al.; our sample of 59 plants across the US had a mean of 15. Note that the central estimate in McDonald et al. (2009) is the midpoint between "reasonable" minimum and maximum values for land-use intensity. For hydroelectric, their min is 1,600 ha/TWh/y and their max is 9,200 ha/TWh/y, which far from captures the full range found in our sample, going from less than 1 ha/TWh/y to over 1 million ha/TWh/y. Smil (2010) offers examples of land-use intensities for a handful of energy sources. For natural gas, he provides a range of about 6–60 ha/TWh/y; our estimated median for natural gas

(footprint) is 410 and for natural gas (spacing), 1900. This large discrepancy could be the result of Smil (2010) basing his estimates on particularly land-efficient natural gas operations, rather than the actual range of existing installations.

For policymakers interested in the land-use effects of expanding a given power source, using our calculated mean LUIE is preferable to using estimates from single cases, since real-world scale-up will include the full distribution of land-use intensities of different installations. Scaling up an energy source means building more of some low-footprint installations and more of some high-footprint installations, so it is the average land-use intensity that will determine the expected overall impacts. Using LUIE estimates from a single installation, which may by chance represent a particularly small or particularly large footprint, may severely bias projected land-use impacts.

Our results suggest that production of electricity to meet decarbonization goals could become a significant new driver of land-use and land-cover change with implications for habitat and biodiversity loss, food security, and other environmental and social priorities. An expanding footprint is not inevitable: the LUIE for integrated PV, nuclear, the footprint of wind, and geothermal are each less than coal or natural gas, which together, currently generate more than 60% of the world's electricity [62].

Impacts of energy development can be mitigated through strategic local-scale approaches that consider proximate impacts within and near development boundaries and landscape-level approaches that target more systemic, cumulative impacts of entire energy systems [85, 86]. Decision-support tools can integrate multiple criteria, leading to reductions in various types of environmental and social impacts while optimizing generation with respect to the cultural and economic interests of stakeholders [87]. Examples of such approaches already exist for several regions and sources, including hydroelectricity and solar energy [69, 80, 88, 89]. However, even with better siting, the larger the aggregate footprint of energy, the more likely environmental impacts are to grow [13]. This underscores the long-term environmental benefits of electricity sources that have both low land and carbon footprints, and the importance of using LUIE as a metric alongside other factors like GHG emissions, cost, and reliability in planning and governance of energy development.

## Supporting information

**S1 Text. Supplementary information.**
(DOCX)

**S1 Table. Details of data sourcing by electricity generation technology.**
(DOCX)

**S1 File. Full dataset of land-use intensity data by fuel source.**
(XLSX)

## Acknowledgments

We thank Crystal Yeh, Matthew MacCaughey, and Madison K. Hoffacker for assistance with data collection, and Laura Small for assistance with data analysis. We also thank Ben Phalan, Patrick Meyfroidt, Ken Caldeira, David Simpson, and Harry Saunders for their helpful comments.

## Author Contributions

**Conceptualization:** Jessica Lovering, Linus Blomqvist.

**Data curation:** Jessica Lovering, Marian Swain, Rebecca R. Hernandez.

**Formal analysis:** Marian Swain.

**Funding acquisition:** Rebecca R. Hernandez.

**Investigation:** Marian Swain.

**Methodology:** Jessica Lovering, Linus Blomqvist, Rebecca R. Hernandez.

**Project administration:** Jessica Lovering.

**Supervision:** Jessica Lovering.

**Validation:** Linus Blomqvist.

**Writing – original draft:** Jessica Lovering, Marian Swain, Linus Blomqvist, Rebecca R. Hernandez.

**Writing – review & editing:** Jessica Lovering, Linus Blomqvist, Rebecca R. Hernandez.

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
