## [Decision Letter · Decision Letter 0]

8 Apr 2022

PONE-D-22-07105Land-use intensity of electricity production and tomorrow’s energy landscapePLOS ONE

Dear Dr. Lovering,

Thank you for submitting your manuscript to PLOS ONE. After careful consideration, we feel that it has merit but does not fully meet PLOS ONE’s publication criteria as it currently stands. Therefore, we invite you to submit a revised version of the manuscript that addresses the points raised during the review process. The authors must respond to all the queries of all the reviewers and the manuscript must be modified as suggested by the reviewer. The paper is recommended for Minor Revision.

We look forward to receiving your revised manuscript.

Kind regards,

Lalit Chandra Saikia, PhD

Academic Editor

PLOS ONE

Journal Requirements:

"R.R.H. received funding for this project from the USDA National Institute of Food and Agriculture Hatch Project 1010512, the Department of Land, Air and Water Resources at UC Davis, and the UC President's Postdoctoral Fellowship."

"R.R.H. received funding for this project from the USDA National Institute of Food and Agriculture Hatch Project 1010512, the Department of Land, Air and Water Resources at UC Davis, and the UC President's Postdoctoral Fellowship.

https://nifa.usda.gov/grants

https://www.lawr.ucdavis.edu/

https://ppfp.ucop.edu/info/ 

Additional Editor Comments:

The authors must respond to all the queries of all the reviewers and the manuscript must be modified as suggested by the reviewer. The paper is recommended for Minor Revision.

Reviewers' comments:

Reviewer's Responses to Questions

**Comments to the Author**

1. Is the manuscript technically sound, and do the data support the conclusions?

Reviewer #1: Yes

Reviewer #2: Partly

Reviewer #3: Yes

2. Has the statistical analysis been performed appropriately and rigorously? 

Reviewer #1: Yes

Reviewer #2: Yes

Reviewer #3: Yes

3. Have the authors made all data underlying the findings in their manuscript fully available?

Reviewer #1: Yes

Reviewer #2: No

Reviewer #3: Yes

4. Is the manuscript presented in an intelligible fashion and written in standard English?

Reviewer #1: Yes

Reviewer #2: Yes

Reviewer #3: Yes

5. Review Comments to the Author

Reviewer #1: With the increase of energy demand, its consumption of land will have impacts on people's health, ecology and environment. This study extensively collects power generation data and land occupancy data of the major electricity sources. Based on these data, the land-use intensity of energy (LUIE) for the main electricity sources, such as coal, natural gas and nuclear energy, is calculated. Calculation results are then applied to future scenarios, which can help policymakers formulate energy development strategies more scientifically. With a large amount of data and detailed analysis, this study provides a good reference for land-use intensity of energy. The paper is recommended for publication in PLOS ONE. Some questions are suggested below.

(1) In the manuscript, the specific method for calculating coat indirect land area is unclear.

(2) The study uses a large amount of data from multiple countries and states in the United States, which undoubtedly increases the authority of the manuscript. However, are the data collected within a similar time period, e.g. recent 5 years or 10 years?

(3) The results of this study suggest that the IEA 6 Degree scenario has the smallest land-use area, which is a business-as-usual scenario. However, the use of a large amount of fossil energy will bring environmental problems. Can the environmental effects be taken into account to calculate and compare the generalized LUIE for various future scenarios, for example, converting the economic loss per unit of carbon emissions into an equivalent increase in land-use area?

(4) The calculation results of LUIE of different studies for the same power source are quite different. This may be attributed to the fact that the amount and weight of data used in each study are different or different calculation standards are used. Using a large amount of data can enhance the accuracy of the calculation results. But how to explain the rationality of the calculation standards chosen in the manuscript?

Reviewer #2: Lovering and colleagues investigate the Land Use Intensity of Electricity (LUIE) production in the United States for a variety of current electricity generation approaches. These approaches include coal, natural gas, nuclear, hydroelectric, biomass, wind, solar photovoltaic, and concentrating solar power (solar-CSP). Given the global need to transition from “business as usual” scenarios for energy production, this is a timely and important study. Moreover, this study is unique in that it considers the impact to land use for each of these approaches which an essential contribution to understanding how electricity generation may compete and further exacerbate resource conflict both here in the United States and abroad as we manage the transition away from fossil fuels. Overall, the manuscript is in good shape. It was a joy to read and provided unique and novel insights which I had not considered. I do have few concerns:

1. For question 1 – “Is the manuscript technically sound, and do the data support the conclusions?” The data as presented are sound and do support the conclusions. I do have one concern which I don’t consider trivial and that is with respect to the storage of spent nuclear fuel. It is true that currently nuclear fuel in the United States is stored onsite. However our stockpile of spent fuel is growing rapidly. For example, An April 2020 report from the Congressional Research Service indicates that the amount of spent nuclear fuel in storage is expected to double by 2048. This is an issue which is unique to nuclear because unlike a hydroelectric dam or natural gas power plant which has a more-or-less fixed footprint at construction, the footprint associated with a nuclear powerplant will continue to grow throughout its lifecycle. I think that if the authors acknowledge this and work it into their paper it likely won’t alter the fundamental findings of the paper and would go a long way to strengthening the paper.

2. For question 2 – “Has the statistical analysis been performed appropriately and rigorously?” Yes, in general. Although I would refer the authors to my comments on item #1.

3. For question 3 – “Have the authors made all data underlying the findings in their manuscript fully available?” I’m not entirely sure how to address this. The compiled data set was generated entirely from publicly available information and the authors do cite each source from which data were obtained. However, the authors do not provide the compiled data set (or, at the very least, I wasn’t able to access it).

4. For question 4 – “Is the manuscript presented in an intelligible fashion and written in standard English?” Yes – the paper was very well written. It was a joy to read.

Reviewer #3: According to the principle of thermodynamics, I am not 100% sure you can write "energy production" or similar. Better to write "energy conversion or management"?

You wrote "To provide LUIE results representative of the current state of each energy technology, we required that data sources represent existing, operational energy facilities and real world, rather than modeled, electricity generation data". What you precisely mean? Did you get the time series data or you got for instance the annual power production for those specific plants from literature (i.e. TWh/year/plant)?

What could be the effect of climate and technological advancements (i.e. increased efficiencies) on your analysis?

Some energy sources, as you have pointed out depends on locations, for instance solar is driven by latitude, could you further discuss on this or even better plot them in a map?

Can you discuss about other emerging technologies? and or provide further suggestions on how to avoid this energy and land use conflict (i.e., marine energy, agrivoltaics, supercritical geothermal, etc.)?

6. PLOS authors have the option to publish the peer review history of their article (what does this mean?). If published, this will include your full peer review and any attached files.

Reviewer #1: **Yes: **DONG　CHEN

Reviewer #2: No

Reviewer #3: **Yes: **Pietro Elia Campana

---

## [Author Response · Author response to Decision Letter 0]

23 May 2022

Per the editor's request, we have removed funding information from the Acknowledgements section of the manuscript. All Reviewer Comments have been addressed as detailed in the attached "Response to Reviewers"

---

## [Decision Letter · Decision Letter 1]

6 Jun 2022

Land-use intensity of electricity production and tomorrow’s energy landscape

PONE-D-22-07105R1

Dear Dr. Lovering,

We’re pleased to inform you that your manuscript has been judged scientifically suitable for publication and will be formally accepted for publication once it meets all outstanding technical requirements.

Kind regards,

Lalit Chandra Saikia, PhD

Academic Editor

PLOS ONE

Additional Editor Comments (optional):

The authors have addressed all the comments of the reviewer. The paper is recommended for publication.

Reviewers' comments:

Reviewer's Responses to Questions

**Comments to the Author**

1. If the authors have adequately addressed your comments raised in a previous round of review and you feel that this manuscript is now acceptable for publication, you may indicate that here to bypass the “Comments to the Author” section, enter your conflict of interest statement in the “Confidential to Editor” section, and submit your "Accept" recommendation.

Reviewer #1: All comments have been addressed

Reviewer #3: All comments have been addressed

2. Is the manuscript technically sound, and do the data support the conclusions?

Reviewer #1: Yes

Reviewer #3: Yes

3. Has the statistical analysis been performed appropriately and rigorously? 

Reviewer #1: Yes

Reviewer #3: Yes

4. Have the authors made all data underlying the findings in their manuscript fully available?

Reviewer #1: Yes

Reviewer #3: Yes

5. Is the manuscript presented in an intelligible fashion and written in standard English?

Reviewer #1: Yes

Reviewer #3: Yes

6. Review Comments to the Author

Reviewer #1: (No Response)

Reviewer #3: All my previous minor comments were addressed. I appreciate the inclusion of each comment response in the text.

7. PLOS authors have the option to publish the peer review history of their article (what does this mean?). If published, this will include your full peer review and any attached files.

Reviewer #1: **Yes: **Dong Chen

Reviewer #3: No

---

## [Editor Report · Acceptance letter]

14 Jun 2022

PONE-D-22-07105R1 

Land-use intensity of electricity production and tomorrow’s energy landscape 

Dear Dr. Lovering:

I'm pleased to inform you that your manuscript has been deemed suitable for publication in PLOS ONE. Congratulations! Your manuscript is now with our production department. 

Kind regards, 

on behalf of

Dr. Lalit Chandra Saikia 

Academic Editor

PLOS ONE